# Elevated Sirtuin 1 Levels in Patients with Chronic Kidney Disease, Including on Peritoneal Dialysis: Associations with Cardiovascular Risk and Peritoneal Fibrosis

**DOI:** 10.3390/ijms26189033

**Published:** 2025-09-17

**Authors:** Angelika Bielach-Bazyluk, Katarzyna Czajkowska, Ewa Koc-Zorawska, Tomasz Hryszko, Edyta Zbroch

**Affiliations:** 1Department of Dermatology and Venereology, Medical University of Bialystok, 15-540 Bialystok, Poland; 2Second Department of Nephrology, Hypertension and Internal Medicine with Dialysis Unit, Medical University of Bialystok, 15-276 Bialystok, Poland; 3Department of Internal Medicine and Hypertension, Medical University of Bialystok, 15-540 Bialystok, Poland

**Keywords:** sirtuin 1, SIRT1, peritoneal dialysis, chronic kidney disease, cardiovascular disease, peritoneal fibrosis

## Abstract

Sirtuin 1 (SIRT1) is implicated in oxidative stress, inflammation, and fibrosis—processes central to chronic kidney disease (CKD) and cardiovascular complications. Increased serum levels of SIRT1 have been reported in dialysis patients, and its role in peritoneal fibrosis, a leading cause of peritoneal dialysis failure, is well established. This study evaluated serum SIRT1 levels in 165 participants: peritoneally dialyzed patients (CAPD), conservatively treated CKD patients (CT), and healthy controls. Serum SIRT1 was measured by ELISA and analyzed alongside clinical factors. SIRT1 concentrations were markedly elevated in CAPD patients compared to both CT patients and controls. In CAPD patients, SIRT1 levels were not influenced by age, sex, dialysis adequacy, residual renal function, or comorbidities, but were higher in those with impaired left ventricular relaxation. Pharmacotherapy affected SIRT1 levels. Multivariate analysis identified phosphate and cholesterol as independent predictors of SIRT1. Our study suggests that serum SIRT1 levels may reflect diverse pathophysiological processes in CKD patients, including those on peritoneal dialysis. Elevated SIRT1 may indicate compensatory mechanisms related to renal dysfunction and cardiovascular stress. Future research on larger, pharmacologically homogeneous groups is warranted to clarify SIRT1’s role in peritoneal fibrosis and its potential as a biomarker of cardiovascular and renal complications in CKD.

## 1. Introduction

Sirtuins constitute a group (SIRT1-7) of enzymes, with predominantly NAD+-dependent deacetylase activity, involved in epigenetic regulation of histone and non-histone proteins [1]. Sirtuins regulate cellular homeostasis in response to metabolic and genotoxic states through several processes, including cell cycle arrest, apoptosis, oxidative stress response, DNA repair, inflammation, and glucose and lipid metabolism. Sirtuin 1 (SIRT1), known as the longevity gene, was found to prolong lifespan and improve numerous age-related diseases in experimental models [2,3,4,5,6]; however, there is no strong evidence for the extension of lifespan in humans so far. The spectrum of SIRT1 activity makes it a significant contributor to the development of cardiovascular disease, chronic kidney disease (CKD), and diabetes.

With regard to kidney function, it alleviates fibrogenesis, maintains podocyte function with a consequent decrease in albuminuria, and reduces blood pressure by the modulation of the renin–angiotensin–aldosterone system (RAAS) and secretion of vasodilatory agents [7,8,9,10,11]. Recently, we have demonstrated that serum SIRT1 concentration is considerably higher in patients with renal failure on maintenance hemodialysis in comparison to healthy subjects [12].

Moreover, numerous lines of evidence show that SIRT1 is implicated in principal complications of renal failure, including coronary artery disease [13], left ventricular hypertrophy [14], and CKD-MBD [15]. These complications are collectively responsible for dramatically increased cardiovascular risk in those patients [16]. On the other hand, SIRT1 is also involved in key molecular pathways underlying peritoneal fibrosis [17,18,19,20], the principal reason for discontinuing peritoneal dialysis [21]. While recent studies have confirmed SIRT1’s role in the pathogenesis of peritoneal fibrosis [22,23,24], it remains unclear whether SIRT1 could serve as a clinically useful biomarker of peritoneal membrane function measurable in serum.

The objective of this study was to assess differences in serum SIRT1 concentrations between CKD patients treated conservatively and those undergoing peritoneal dialysis. Additionally, we evaluated potential correlations between SIRT1 levels and peritoneal membrane function (assessed by the Kt/V index), demographic characteristics, clinical parameters, comorbidities such as cardiovascular disease and diabetes, and pharmacological treatments.

## 2. Results

### 2.1. Studied Population

In total, 140 serum samples from patients enrolled in the study and 25 samples from controls were analyzed. The study population with CKD was stratified into two groups: those undergoing continuous ambulatory peritoneal dialysis (CAPD) and those receiving conservative treatment (CT). The CT subgroup comprised 51 females and 49 males, with a mean age of 74 years (range: 23–91). The median age of the CAPD subgroup was 54 years (range 21–80) and comprised 25 (63%) women. The subgroups differed with respect to age, body mass index (BMI), laboratory measurements, and pharmacological treatment. The specific characteristics of patients included in the study are summarized in Table 1.

### 2.2. Comparison of the Sirtuin Level in Chronic Kidney Disease Patients and the Control Group

Median sirtuin 1 concentration differed considerably in the studied subgroups and in both was significantly higher than in the control group (9.61 vs. 2.23 vs. 0.44 ng/mL, *p* < 0.001); see Figure 1.

Median sirtuin 1 level in patients undergoing peritoneal dialysis was over 4-fold higher than in conservatively treated patients (9.61 [6.65–19.45] vs. 2.23 [1.57–3.24] ng/mL, *p* < 0.001). We observed approximately a 5-fold increase in median sirtuin 1 concentration between the control group (0.44 [0.19–0.64] ng/mL) and CT and over a 20-fold increase compared to the CAPD subgroup.

### 2.3. Sirtuin Level in CAPD Patients and Its Associations with Selected Variables

Basic characteristics of the CAPD subgroup are presented in Table 2.

The dialysis vintage was 32 (18–54) months. Sirtuin concentration was not influenced by age, sex, residual renal function, underlying etiology of renal failure, treatment with erythropoietin, or dialysis adequacy as measured by kt/V. The presence of hypertension, diabetes mellitus, or New York Heart Association (NYHA) class of heart failure was not associated with sirtuin level.

Patients with impaired left ventricular relaxation had higher SIRT1 concentrations (Me = 14.67 [7.46–21.28] vs. Me = 7.75 [5.39–9.25] ng/mL, *p* < 0.05). SIRT1 concentration was weakly correlated with left ventricular ejection fraction (R = −0.34, *p* < 0.05).

### 2.4. The Relationship Between Sirtuin 1 Level and Selected Variables in the Whole Studied Group

The analysis of the two subgroups taken together revealed several associations with clinical measurements (see Table 3). SIRT1 concentration was associated with serum phosphates (R = 0.7, *p* < 0.001), age (R = −0.37, *p* < 0.001), BMI (R = −0.29, *p* < 0.001), mean diastolic blood pressure (mDBP) (R = 0.31, *p* < 0.001), iPTH (R = 0.4, *p* <0.001), ferritin (R = 0.3, *p* < 0.01), and total cholesterol (R = 0.26, *p* < 0.01). Patients with concomitant diabetes had significantly lowered serum SIRT1 concentrations (Me = 2.4 [1.6–4.8] vs. Me = 3.3 [2.1–7.4] ng/mL, *p* < 0.05).

Regarding differences related to pharmacological treatment, use of statins, α1-receptor blockers, and diuretics was associated with a decrease in SIRT1 level (Me = 2.2 [1.6–5.1] vs. 3.5 [2.2–5.4], *p* < 0.05; Me = 2.3 [1.6–3.6] vs. Me = 3.1 [1.9–7.4], *p* < 0.05; Me = 2.4 [1.6–3.8] vs. Me = 5.1 [2.7–9.0] ng/mL, *p* < 0.001, respectively), whereas the level was increased in angiotensin-converting enzyme inhibitor (ACEi) users (Me = 3.3 [2.2–8.4] vs. 2.5 [1.6–4.9] ng/mL, *p* < 0.05).

Patients with impaired left ventricular relaxation had higher SIRT1 concentrations (Me = 5.6 [2.2–15.0] vs. Me = 2.6 [1.8–5.1] ng/mL, *p* < 0.01). The increase in right ventricle size assessed on echocardiography correlated inversely with SIRT1 concentration (R = −0.24, *p* < 0.05).

All significantly correlated variables were included in multivariate multiple regression models to confirm which of those have an independent association. After adjustment, only phosphates (β = 0.52, *p* = 0.001) and total cholesterol level (β = 0.42, *p* < 0.05) were independent predictors associated with serum SIRT1 level, and the model explained 55% of its variation among patients with CKD (*p* < 0.05), as shown in Table 4.

## 3. Discussion

### 3.1. The Emerging Role of Sirtuin 1 in Renal Pathophysiology and Clinical Implications in CKD

With regard to kidney function, a growing body of evidence from in vitro and animal studies has confirmed the crucial role of SIRT1 in renal diseases [8,11,25,26,27]. Experimental models have demonstrated that SIRT1 reduces albuminuria by preserving podocyte function [8], reduces oxidative stress [26], modulates the advanced glycation end products and receptor for advanced glycation end products (AGEs-RAGE) axis [27], and attenuates renal fibrosis [11,25]. However, despite these extensive preclinical findings, only a limited number of studies have assessed serum SIRT1 concentrations in patients with CKD. We previously reported that serum SIRT1 levels are significantly elevated in CKD patients, both those receiving conservative [8,11,25,26,27] management and those undergoing maintenance hemodialysis, with concentrations inversely correlated with renal function and peaking in end-stage disease [12,28]. Furthermore, a recent study demonstrated increased SIRT1 levels in kidney transplant recipients, where higher concentrations were associated with cardiovascular events [29]. The data support the notion that serum SIRT1 may serve a dual role as a biomarker for CKD and as a prognostic indicator of cardiovascular risk.

CAPD is typically recommended for patients with end-stage renal disease who seek a home-based dialysis modality, especially those who are either unable to undergo hemodialysis or prefer CAPD due to lifestyle considerations. Although it may represent a more convenient form of treatment for some patients compared to hemodialysis, this type of therapy is associated with specific complications, including peritoneal fibrosis, which forces the cessation of therapy and the transition to hemodialysis [30].

In relation to CAPD, recent findings in diabetic patients revealed that SIRT1 concentration is linked to filtration membrane efficiency and negatively correlates with creatinine clearance [31]. To the best of our knowledge, this research represents the first direct comparison of serum SIRT1 levels between conservatively managed CKD patients and those undergoing CAPD.

### 3.2. Mechanistic Insights into Peritoneal Fibrosis with a Focus on SIRT1

#### 3.2.1. Pathogenesis of Peritoneal Fibrosis

The peritoneum serves as a natural semipermeable membrane that facilitates ultrafiltration and solute diffusion during dialysis [32]. Peritoneal fibrosis remains a significant clinical challenge, with its pathogenesis the focus of numerous investigations [19,33,34,35,36]. The underlying mechanisms are multifactorial, involving chronic inflammation induced by repeated exposure to bioincompatible peritoneal dialysis fluids, macrophage activation, mesothelial-to-mesenchymal transition (EMT), neovascularization, and oxidative stress [17,18,20,37]. The composition of dialysis fluid is particularly important, as glucose-rich solutions promote chronic inflammation and fibrotic changes [38,39,40]. SIRT1 has been shown to protect against oxidative stress by enhancing mitochondrial biogenesis and activating antioxidant enzymes [41,42]. Notably, oxidative stress induced by non-physiological peritoneal dialysis fluid can be mitigated through the AMP-activated protein kinase/SIRT1 (AMPK/SIRT1) pathway, as demonstrated with the addition of caffeic acid phenethyl ester (CAPE) [22].

#### 3.2.2. The Anti-Fibrotic Role of SIRT1

At the molecular level, SIRT1 exerts antifibrotic effects through multidirectional regulation of gene expression and protein activity [43,44]. A key mechanism involves downregulation of transforming growth factor β (TGF-β) signaling via Smad3 deacetylation, a pathway relevant to fibrosis in multiple organs [45,46,47,48,49,50,51,52,53,54]. In a recent mouse model of peritoneal fibrosis, Guo et al. [24] demonstrated that SIRT1 overexpression ameliorates fibrosis, whereas SIRT1 knockout mice exhibited aggravated fibrotic changes compared to wild-type controls. Furthermore, the same group reported that human umbilical cord mesenchymal stem cells (hUCMSCs) overexpressing SIRT1 display therapeutic potential in peritoneal fibrosis, as administration of SIRT1-modified hUCMSCs enhanced antifibrotic efficacy, inhibited EMT, and improved ultrafiltration [23].

Beyond TGF-β/Smad signaling, SIRT1 mitigates inflammation through deacetylation of the NF-κB p65 subunit, thereby reducing the expression of tumor necrosis factor α (TNF-α) and interleukin 6 (IL-6), both critical mediators of EMT and peritoneal fibrosis [55,56,57,58,59,60]. Notably, SIRT1 activators have been shown to inhibit EMT, further reinforcing its protective role and highlighting the therapeutic potential of SIRT1 in peritoneal fibrosis [61].

#### 3.2.3. Macrophage Polarization, Neovascularization, and SIRT1 in Fibrosis

SIRT1 is also critically involved in regulating macrophage polarization, balancing pro-inflammatory M1 macrophages and anti-inflammatory M2 macrophages, which are engaged in tissue repair [62,63,64]. SIRT1 promotes M2 and the resolution of inflammation [65,66,67,68]. Although M2 macrophages play a pivotal role in tissue repair, their excessive polarization may exacerbate peritoneal fibrosis by promoting fibroblast proliferation and extracellular matrix accumulation [69]; however, direct evidence in peritoneal dialysis patients remains limited.

In the context of peritoneal fibrosis, SIRT1 has been implicated in neoangiogenesis through modulation of vascular endothelial growth factor (VEGF) production, potentially intensifying fibrosis [18,70]. However, the precise role of SIRT1 in neoangiogenesis remains under investigation, as it may either promote or inhibit new vessel formation depending on the stage and severity of fibrosis.

### 3.3. SIRT1 as a Potential Biomarker of Peritoneal Filtration Failure in Dialysis Patients

It remains an open question whether the concentration or activity of SIRT1 measured in the blood serum of patients undergoing peritoneal dialysis can be used as a marker of the risk of peritoneal filtration failure. Preliminary findings suggest that a decline in SIRT1 levels is associated with reduced dialysis adequacy and impaired peritoneal transport function [31]. Specifically, a study conducted on a group of 140 patients with diabetic nephropathy provides supporting evidence, as it demonstrated that SIRT1 expression in the serum decreases in parallel with Kt/V, a standard indicator of dialysis adequacy [31]. Moreover, SIRT1 expression was inversely associated with creatinine clearance and CA125 expression, a glycoprotein frequently utilized to assess peritoneal transport capacity and indicative of mesothelial cell mass [31,71]. These observations underscore the need to develop novel clinical markers to facilitate early identification of individuals susceptible to filtration failure, enabling timely planning for the transition to hemodialysis or preparation for kidney transplantation.

### 3.4. SIRT1 in Calcium-Phosphate Metabolism

Beyond its role in fibrosis, SIRT1 is also implicated in phosphate metabolism, vascular calcification, and mineral-bone disorders in CKD. Molecular studies in animal models have shown that a milieu rich in glucose and phosphates downregulates SIRT1 expression, activating signaling pathways that contribute to vascular calcification—a hallmark of aging and conditions linked to elevated cardiovascular risk [72]. In our study, serum SIRT1 concentration was inversely correlated with age and BMI, and positively associated with phosphates and iPTH. Of note, SIRT1 concentration in the CAPD group was four times greater than in the CT group, which was statistically older, and remarkably higher than in the age-matched control cohort. Furthermore, multivariate analysis identified only phosphate and total cholesterol concentration as independent predictors of SIRT1 concentrations. The findings suggest that disturbances in calcium-phosphate metabolism have a greater impact on SIRT1 levels than demographic factors. Emerging evidence also supports the potential of SIRT1 as a clinical marker of osteoporosis risk, as shown in a study involving patients with newly diagnosed type 2 diabetes [73]. More research is needed to fully elucidate both the direct and indirect effects of SIRT1 on serum phosphate levels and phosphate-related diseases, including CKD-MBD.

### 3.5. SIRT1 in Lipid Metabolism

SIRT1 has been extensively reported to influence lipid metabolism through the modulation of key regulatory proteins, including sterol regulatory element-binding protein (SREBP), AMP-activated protein kinase (AMPK), peroxisome proliferator-activated receptor gamma (PPARγ), as well as the nuclear liver X receptor (LXR) and farnesoid X receptor (FXR), underscoring its central role in coordinating multiple pathways involved in lipid homeostasis [74]. By deacetylating various transcription factors, SIRT1 integrates signals controlling lipid synthesis, storage, and efflux, positioning it as a pivotal regulator of lipid metabolism [74]. Mechanistically, SIRT1 regulates lipid and cholesterol metabolism by deacetylating LXRs to promote ATP-binding cassette transporter A1 and G1 (ABCA1/ABCG1)-mediated cholesterol efflux and repressing SREBP-1c-driven lipogenesis [75,76]. It also interacts with FXR-dependent bile acid signaling [77], enhances PPARγ activity to stimulate fatty acid oxidation [78], and activates AMPK [79], which further inhibits lipogenesis. In previous models, reduced SIRT1 expression has been associated with impaired reverse cholesterol transport [75], increased atherosclerotic risk [80], and susceptibility to gallstone formation [81], whereas enhanced SIRT1 activity exerts protective effects against non-alcoholic fatty liver disease (NAFLD) [82]. In humans, SIRT1 has also been linked to improved HDL functionality and favorable lipid profiles [83,84,85]. Other investigations have demonstrated a positive correlation between circulating SIRT1 concentrations and triglyceride levels. Furthermore, interventional trials with the pharmacological activator SRT2104 indicate that SIRT1 modulation can influence lipid profiles; however, the effects appear context-dependent: some studies demonstrated reductions in cholesterol and triglycerides, while others found neutral outcomes. These discrepancies likely reflect differences in study populations, dosage regimens, and treatment duration, underscoring the complexity of SIRT1’s role in lipid homeostasis [86,87]. Consistent with this, our study shows that circulating SIRT1 levels are significantly associated with serum cholesterol concentrations, further supporting its potential role in systemic cholesterol homeostasis. Interestingly, whereas previous studies have generally reported inverse associations between SIRT1 expression and circulating cholesterol levels, our data reveal a positive correlation between circulating SIRT1 and total serum cholesterol, suggesting that the relationship between SIRT1 and systemic cholesterol homeostasis may be context-dependent and warrants further investigation.

### 3.6. SIRT1 Concentration in Diabetic Patients

Our study demonstrated that individuals suffering from diabetes exhibited lower SIRT1 concentrations compared with CKD patients without concomitant diabetes. The concentration of sirtuin 1 in patients with diabetes has been extensively investigated in previous studies [88,89,90,91,92,93,94]. Most large-scale studies suggest that serum SIRT1 is diminished in type 2 diabetic patients, regardless of concomitant diabetic nephropathy [88,89]. Particularly low concentrations of SIRT1 have been noticed in patients with poorly controlled diabetes [89,90]. Although patients with diabetic kidney disease tend to exhibit lower SIRT1 levels, findings on the correlation between SIRT1 and the severity of kidney damage remain inconsistent [88,89,91,92,93,94]. In a study involving nearly 700 patients, a significant relationship was found between SIRT1 concentration and the albumin-to-creatinine ratio [88]; however, other studies failed to confirm any association between SIRT1 levels and either the degree of albuminuria or estimated glomerular filtration rate (eGFR) [89,91]. Conflicting findings have emerged from several smaller studies, some of which showed greater SIRT1 concentrations in diabetic patients than in healthy controls [92,93]. Notably, a study involving young women with type 1 diabetes reported SIRT1 levels similar to those observed in healthy individuals [94].

### 3.7. SIRT1 and Cardiovascular Risk Among At-Risk Patients

Another area of research concerning serum SIRT1 concentration explores its potential as a reliable marker of cardiovascular risk. A study involving patients with diabetic kidney disease, divided into two groups based on the presence of coronary artery disease, found no significant differences in serum SIRT1 levels between the groups [95]. This finding contrasts with other studies suggesting that lower serum SIRT1 may be associated with elevated cardiovascular risk [96,97]. For instance, a recently published prospective observational study in patients with diabetic kidney disease undergoing lower limb revascularization identified low SIRT1 levels as an independent risk factor for both major adverse cardiovascular events (MACE) and major adverse limb events (MALE). Moreover, incorporating this parameter into existing risk assessment models significantly improved the accuracy of forecasting adverse outcomes [96]. Further supporting evidence comes from a cross-sectional study involving asymptomatic individuals with low to intermediate Framingham Risk Scores [97]. Within this cohort, lower serum SIRT1 concentrations were independently associated with the presence of high-risk coronary plaques as detected by coronary computed tomography angiography (CTA). Collectively, these findings suggest that serum sirtuin 1 may have value as a cardiovascular risk biomarker; however, more research is needed to establish its clinical utility.

### 3.8. The Impact of Antihypertensive and Lipid-Lowering Therapies on SIRT1 Levels

Regarding pharmacotherapy, lower serum SIRT1 concentrations were observed in patients receiving statins, α-1 blockers, or diuretics. In contrast, higher SIRT1 levels were found in patients receiving ACEi. To date, published scientific data on the effects of these medications on serum SIRT1 levels in humans are available only for statins [98,99,100]. Evidence suggests that statins may modulate SIRT1 expression, with most studies reporting upregulation at the gene or protein level [99,100]. However, some studies involving patients with coronary artery disease have reported downregulation of SIRT1, implying that statins may normalize elevated baseline SIRT1 expression [98].

### 3.9. SIRT1 in Ventricular Dysfunction

Our findings indicate that altered SIRT1 concentrations may reflect distinct pathophysiological processes in patients with cardiac dysfunction. Specifically, elevated SIRT1 levels observed in individuals with impaired left ventricular relaxation—as evidenced by echocardiographic parameters—may represent a compensatory response to increased afterload. This condition is frequently driven by systemic arterial hypertension and may be further exacerbated by vascular stiffness and calcification secondary to disturbances in calcium-phosphate metabolism. The development of heart failure is closely linked to mitochondrial dysfunction, with SIRT1 modulating the maladaptive metabolic response of cardiomyocytes under conditions of pressure overload [101]. In contrast, decreased SIRT1 concentrations in patients with right ventricular hypertrophy may be attributed to chronic volume overload, which is common in advanced cardiorenal syndromes. These reductions may also be influenced by hemodilution due to fluid retention. Another potential explanation is the distinct embryological origin of the right ventricle, which may contribute to electrophysiological differences between right and left ventricles observed in heart failure [102,103]. Additionally, it has been shown that resveratrol supplementation can reverse right ventricular cardiomyocyte dysfunction secondary to pulmonary hypertension, in contrast to no significant histological changes noted in the left ventricle [104].

### 3.10. Hypothesizing a Compensatory Role of SIRT1 in Cardiorenal Dysfunction

We propose that elevated SIRT1 concentrations in this context may reflect compensatory mechanisms aimed at counteracting reduced enzymatic activity. Further studies are warranted to examine the correlation between sirtuin concentration and activity in more homogeneous cohorts, employing specific markers of glomerular filtration and linking these findings with inflammatory mediators, including TNF-α and IL-6. This approach may offer deeper insights into the involvement of sirtuins in the cardiorenal network and their possible application as clinical markers or potential therapeutic avenues.

## 4. Materials and Methods

### 4.1. Study Design

The study was designed as an observational, cross-sectional investigation, in which patients were consecutively enrolled. Assignment to the CT or CAPD subgroup was determined by the natural progression of the underlying renal disease and the clinical judgment of the treating physicians, rather than by any study-related allocation. No study-related interventions influenced group assignment.

### 4.2. Study Population

A total of 140 adult patients with chronic kidney disease were enrolled and allocated to two subgroups: those receiving conservative treatment (CT, n = 100) and those undergoing peritoneal dialysis (CAPD, n = 40). The control group comprised 25 patients referred to the nephrology outpatient clinic for evaluation of suspected renal dysfunction, in whom chronic kidney disease was subsequently excluded. Additional eligibility criteria for the control group included the absence of clinically manifest cardiovascular disease and no history of pharmacological treatment at the time of enrollment. Patients with acute kidney injury, those undergoing hemodialysis, and renal transplant recipients were excluded from the study.

### 4.3. Data Collection

Clinical data were obtained from patients’ medical history, physical examination, and routine laboratory tests conducted during hospitalization or ambulatory visits. Blood samples for measuring serum sirtuin 1 (SIRT1) concentration were collected once, either during hospitalization for the CT subgroup or during routine ambulatory visits for CAPD patients and controls. SIRT1 levels were quantified using a commercial ELISA kit employing a monoclonal antibody specific to sirtuin 1 (EIAab Science Co., Wuhan, China; limit of detection 32 pg/mL; intra- and inter-assay precision 4.3% and 7.2%, respectively). The reference range for SIRT1 was 78–5000 pg/mL. All assays were performed according to the manufacturer’s instructions.

Blood pressure (BP) was measured in the sitting position using an automatic manometer. The arithmetic mean of three measurements taken during different ambulatory examinations was used for analysis. Well-controlled blood pressure was defined according to the Polish Society of Hypertension/Polish Cardiac Society Expert guidelines and K/DOQI recommendations, as systolic BP < 120 mmHg [105,106].

For CAPD patients, peritoneal dialysis adequacy was assessed using Kt/V and the peritoneal equilibration test (PET), according to standard methods. Residual renal function was evaluated based on 24 h urine collection, with function defined as urine output greater than 100 mL over a 24 h period.

### 4.4. Statistical Analysis

All statistical analyses were conducted using Statistica version 13.1 (Dell Inc., Round Rock, TX, USA) and R version 3.3.3 (Vienna, Austria). Continuous variables are presented as medians with interquartile ranges (IQR), while categorical variables are expressed as frequencies and percentages. The Shapiro–Wilk test was employed to assess the normality of data distribution. Differences between the groups were analyzed using either the Student’s *t*-test or the Mann–Whitney U test, based on the distribution of the data. A two-tailed *p*-value of less than 0.05 was considered statistically significant. Post hoc calculation of effect size (Cohen’s d > 1.5) and power estimates indicated that the study had sufficient statistical power to detect the observed differences. Multivariable conditional logistic regression was used to investigate the association between the covariates and sirtuin 1 levels. The Kruskal–Wallis non-parametric test was used to evaluate sirtuin 1 concentrations in relation to patients’ age across groups with different etiologies of renal failure. The distribution of sex across the study groups was evaluated using a chi-square test.

### 4.5. Bioethical Considerations and Equipment

All enrolled patients provided written informed consent, and the study protocol was approved by the Ethics Committee of the Medical University in Bialystok (approval number: R-I-002/455/2016). The study adhered to the principles outlined in the Helsinki Declaration.

The research was supported with the equipment purchased through the Medical University of Bialystok under the RPOWP 2007–2013 funding program, Priority I, Axis 1.1, contract No. UDA-RPPD.01.01.00-20-001/15-00, dated 26 June 2015.

## 5. Conclusions

Our study highlights the potential role of serum SIRT1 as a biomarker reflecting various pathophysiological processes in patients with CKD, including those undergoing CAPD. Elevated SIRT1 levels in CKD patients may indicate a compensatory response to renal dysfunction, disturbances in mineral metabolism, and cardiovascular stress. Conversely, lower SIRT1 concentrations observed in diabetic patients and those with right ventricular hypertrophy suggest a complex, and context-dependent regulation of this protein. Importantly, multivariate analysis identified phosphate and total cholesterol as independent predictors of SIRT1 levels, further supporting the link between SIRT1 and disturbances in mineral and lipid metabolism in CKD.

SIRT1 appears to be involved in key mechanisms underlying peritoneal fibrosis, inflammation, and oxidative stress. Its expression may also correlate with filtration membrane function in CAPD patients, potentially serving as an early marker of peritoneal failure risk. Moreover, pharmacotherapy, including the use of statins and ACEi, appears to influence circulating SIRT1 levels.

This study has several limitations. The relatively small sample size and heterogeneity of the study population may limit the generalizability of the findings. The sample consisted of consecutive patients hospitalized in the Department of Nephrology, and allocation to the CT or CAPD subgroup reflected the natural course of the underlying kidney disease and clinical decisions, rather than study-driven assignment. No a priori sample size calculation was performed, and the cross-sectional design precludes conclusions about causality. Additionally, the inclusion of patients receiving medications known to modulate SIRT1 could have influenced the observed associations. These factors should be considered when interpreting the findings.

More evidence is needed to establish the relationship between SIRT1 concentration and activity, its role in the progression of peritoneal fibrosis, and its potential as a biomarker for both cardiovascular and renal complications in CKD patients. Future studies should include larger, more homogeneous patient populations, carefully account for confounding factors, and incorporate prospective sample size calculations to better define the clinical relevance of SIRT1 as a biomarker in CKD.

## Figures and Tables

**Figure 1 ijms-26-09033-f001:**
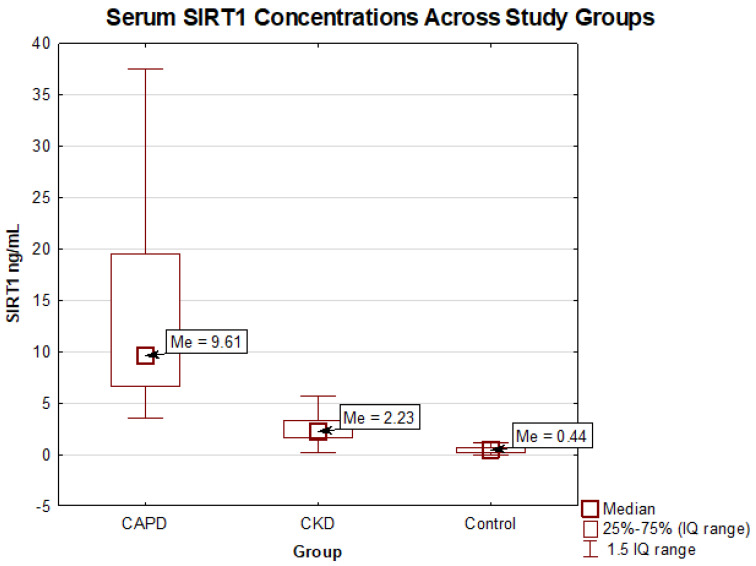
The comparison of serum sirtuin 1 concentrations between studied groups. Abbreviations: SIRT1—sirtuin 1; CAPD—chronic ambulatory peritoneal dialysis; CKD—chronic kidney disease conservatively treated group. Differences between groups were tested with Mann–Whitney U test.

**Table 1 ijms-26-09033-t001:** Comparison of the CAPD and CT subgroups. Continuous variables are presented as median (interquartile range), and categorical variables are expressed as frequency and percentage. Group comparisons were performed using Student’s *t*-test or the Mann–Whitney U test, while sex distribution was assessed using the Chi-square test. A *p*-value < 0.05 was considered statistically significant.

	CAPD	CT	*p* Value
Sirtuin 1 (ng/mL)	9.61 (6.65–19.45)	2.23 (1.57–3.24)	<0.001
Age	54 (46–67)	74 (62.5–80)	<0.001
Gender (vs. male)	62.5% female N = 25	51% N = 51	0.29
BMI (kg/m^2^)	24 (22.2–28.1)	27.7 (24.8–32.9)	0.002
HGB (mg/dL)	11.9 (11.2–12.8)	10.8 (9.9–12.1)	0.004
Calcium (mmol/L)	2.17 (2.01–2.25)	2.23 (2.1–2.31)	0.11
Phosphate (mmol/L)	5.84 (4.38–6.76)	1.22 (1.03–1.5)	<0.001
iPTH (pg/mL)	331.9 (171–626)	146.5 (88.1–243.6)	<0.001

Abbreviations: CAPD—continuous ambulatory peritoneal dialysis; CT—conservatively treated subgroup; BMI—body mass index; HGB—hemoglobin; iPTH—intact parathormone.

**Table 2 ijms-26-09033-t002:** Basic characteristics of the CAPD group.

Variable	Value
Age (yrs)	57 (46–67)
Sex (M/F)	15/25
Dialysis vintage (months)	32 (18–54)
Diabetes mellitus n (%)	14 (35)
HGB (g/dL)	11.85 (11.20–12.75)
ESA treatment n (%)	32 (80)
Albumin (g/dL)	3.54 (3.23–3.85)
kt/V	2.00 (1.70–2.29)
Residual renal function (ml)	350 (100–900)

Abbreviations: ESA—erythropoiesis-stimulating agent; HGB—hemoglobin; kt/V—k (clearance of urea, mL/min), t (duration of the dialysis session, min), V (volume of distribution of urea, L).

**Table 3 ijms-26-09033-t003:** Correlations between SIRT1 concentrations and clinical measurements. Partial correlations were calculated using Spearman’s rank correlation or Pearson’s correlation, following natural logarithm transformation to approximate a normal distribution.

	CAPD + CT	
	R	*p* Value
Age [ln]	−0.37	<0.001
Gender (vs. male)	-	0.71
Body mass index (kg/m^2^) [ln]	−0.29	<0.001
HGB (mg/dL)	0.17	0.05
Calcium (mmol/L)	−0.07	0.43
Pi (mmol/L) [ln]	0.7	<0.001
iPTH (pg/mL)	0.4	<0.001
Glucose (mg/dL)	−0.02	0.79
Total cholesterol	0.26	0.003
Triglycerides (mg/dL)	0.09	0.34
mSBP (mmHg)	0.04	0.64
mDBP (mmHg)	0.31	<0.001
Iron (µg/dL)	0.07	0.46
Ferritin (ng/mL)	0.3	0.002
TIBC (µg/dL)	−0.09	0.51

Abbreviations: HGB—hemoglobin; Pi—phosphates; iPTH—intact parathyroid hormone; mSBP—mean systolic blood pressure; mDBP—mean diastolic blood pressure; TIBC—total iron-binding capacity.

**Table 4 ijms-26-09033-t004:** Multivariate regression analysis of factors associated with SIRT1 concentrations.

Regression Summary R = 0.74, R^2^ = 0.55; F = 3.8, *p* < 0.0						
**Variable**	**Β ***	**SE of β ***	**β**	**SE of β**	**t**	** *p* **
Ln(Age)	−0.241008	0.175360	−1.24207	0.903745	−1.37436	0.181525
Ln(BMI)	0.056816	0.152665	0.53827	1.446358	0.37216	0.712911
mDBP	0.082300	0.163090	0.01016	0.020142	0.50463	0.618236
Ln(Pi)	0.521498	0.145978	1.18510	0.331733	3.57245	0.001472
iPTH	0.040628	0.145586	0.14879	0.533197	0.27906	0.782493
Cholesterol	0.425150	0.157242	0.01376	0.005088	2.70379	0.012151
Ferritin	0.043267	0.158618	0.00067	0.002448	0.27277	0.787268
Right Ventricle Size	0.132423	0.146562	0.04121	0.045615	0.90353	0.374871

* Abbreviations: Ln—natural logarithm; iPTH—intact parathyroid hormone; Pi—phosphate; mDBP—mean diastolic blood pressure; β—standardized coefficient; β—unstandardized coefficient. Multiple regression analysis with Student’s *t*-test for coefficients and F-test for overall model significance.

## Data Availability

The original contributions presented in this study are included in the article. Further inquiries can be directed to the corresponding authors.

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
