# Peer review of "Elevated Sirtuin 1 Levels in Patients with Chronic Kidney Disease, Including on Peritoneal Dialysis: Associations with Cardiovascular Risk and Peritoneal Fibrosis"

_ijms, 2025, doi:10.3390/ijms26189033_

Round 1
Reviewer 1 Report
Comments and Suggestions for Authors
The authors evaluate the association between sirtuin 1 and peritoneal fibrosis and cardiovascular risk in two groups of CKD patients, one receiving conservative treatment and the other receiving peritoneal dialysis. The study is well conducted, and the methods are adequate. However, I have some observations regarding the manuscript and the way in which the information is presented.
1. The abstract does not mention that cholesterol and phosphates were found to be independent predictors, and this is an important finding that should be included in the abstract. I suggest restructuring the abstract.
2. The materials and methods section should be placed after the introduction and before the results section for a better understanding of the manuscript. It should also be divided into sections such as study participants, data collector, statistical analysis, bioethical considerations, etc.
3. How was this sample size calculation arrived at? What was the margin of error for the sample size? 4. How was the variance inflation analysis evaluated? Please include the VIF values.
5. Review the format, size, and font of the tables to make them uniform. Also, in the tables, include in the table footnotes what statistical tests were used to evaluate these variables.
6. Place a table with the variables and significance of the logistic regression.
7. The discussion section should be reorganized because, although it has relevant data, it gives the impression that it is a narrative review article and is too extensive. It should focus on the results found in this study and the relationship between sirtuin 1 and cardiovascular risk and peritoneal fibrosis. In addition, the importance of phosphate and cholesterol as independent predictors should be addressed in more detail, for example, the possible effect of sirtuin 1 on cholesterol metabolism (for example, the interaction with the farnesoid receptor, the liver X receptor, sterol response regulatory elements, etc.). Include comparisons with similar studies.
8. The conclusion should include: Include relevant findings (cholesterol and phosphates as independent predictors).
9. There are some minor grammatical and syntax errors throughout the article and in one of the tables. Please review them.
Best regards.
Author Response
RESPONSE TO REVIEWER 1 COMMENTS
We sincerely appreciate your time and effort in reviewing this manuscript. Below, we provide detailed responses to your comments, with all corresponding revisions and corrections clearly highlighted in the re-submitted files.
- The abstract does not mention that cholesterol and phosphates were found to be independent predictors, and this is an important finding that should be included in the abstract. I suggest restructuring the abstract.
Thank you very much for this valuable comment. We would like to kindly point out that this information is already included in the abstract: “Multivariate analysis identified phosphate and cholesterol as independent predictors of SIRT1.” To make it more visible, we have now highlighted this part in yellow.
- The materials and methods section should be placed after the introduction and before the results section for a better understanding of the manuscript. It should also be divided into sections such as study participants, data collector, statistical analysis, bioethical considerations, etc.
Thank you for your valuable comment. The order of the sections in the manuscript was prepared according to the official article template provided on the journal’s website. Therefore, we would like to leave the final decision regarding moving the Materials and Methods section before the Results section to the editors of IJMS. The Materials and Methods section has been divided into shorter paragraphs in accordance with your recommendation.
How was this sample size calculation arrived at? What was the margin of error for the sample size?
Thank you for raising this point. No a priori sample size calculation was performed, as this study was based on a consecutive sample of patients hospitalized in the Department of Nephrology during the study period. Patients were allocated to the respective groups (conservative management vs. peritoneal dialysis) according to the natural course of their underlying kidney disease and clinical decisions made by the treating physicians, rather than by any study-related assignment. The sample size therefore reflects the number of patients who met the predefined inclusion criteria and fell into each group as a result of routine clinical care, rather than a prospectively calculated target.
Accordingly, a formal margin of error for the sample size was not established. The study was designed as exploratory and hypothesis-generating, utilizing all available eligible patients during the study period.
We recognize this as a limitation and have emphasized in the revised manuscript that future prospective studies should include an a priori sample size calculation to provide a predefined margin of error based on the expected effect sizes.
We would also like to emphasize that the comparison is statistically justified given the very large observed differences in median SIRT1 levels between groups. Post hoc effect size estimates (Cohen’s d > 1.5) and corresponding power calculations indicated that the study had sufficient power to detect these differences, confirming that the results are robust.
- How was the variance inflation analysis evaluated? Please include the VIF values.
The variance inflation analysis was conducted in Statistica. As part of the regression diagnostics, the Tolerance statistic was computed for each predictor, from which the corresponding Variance Inflation Factor (VIF) values were derived (VIF = 1/Tolerance). In our analysis, all VIF values fell within the range 1.17–1.72, thus well below the conventional threshold of 5, indicating no problematic multicollinearity.
The table below presents the obtained values of Tolerance and VIF for each predictor:
|
Variable |
VIF |
Tolerance |
|
Ln(Age) |
1,72 |
0,586687 |
|
Ln(BMI) |
1,30 |
0,774083 |
|
DBP |
1,47 |
0,678285 |
|
Ln(Pi) |
1,18 |
0,846630 |
|
iPTH |
1,17 |
0,851191 |
|
Cholesterol |
1,37 |
0,729670 |
|
Ferritin |
1,39 |
0,717073 |
|
Right Ventricle Size |
1,19 |
0,839887 |
- Review the format, size, and font of the tables to make them uniform. Also, in the tables, include in the table footnotes what statistical tests were used to evaluate these variables.
We have carefully reviewed all tables and ensured that the format, size, and font are now uniform throughout the manuscript. Additionally, we have added footnotes to each table specifying the statistical tests used to evaluate the respective variables, in accordance with your recommendation.
6. Place a table with the variables and significance of the logistic regression.
Thank you for your suggestion. We have added a table presenting the variables included in the logistic regression along with their coefficients, standard errors and p-values.
7. The discussion section should be reorganized because, although it has relevant data, it gives the impression that it is a narrative review article and is too extensive. It should focus on the results found in this study and the relationship between sirtuin 1 and cardiovascular risk and peritoneal fibrosis. In addition, the importance of phosphate and cholesterol as independent predictors should be addressed in more detail, for example, the possible effect of sirtuin 1 on cholesterol metabolism (for example, the interaction with the farnesoid receptor, the liver X receptor, sterol response regulatory elements, etc.). Include comparisons with similar studies.
We thank the reviewer for this comment. We agree that the primary focus of the Discussion should be on relating our findings to the existing body of knowledge. At the same time, our intention was to place the results within a broader pathophysiological framework of chronic kidney disease to highlight the potential role of SIRT1 as a biomarker. We believe that this broader perspective helps contextualize the observed associations more effectively.
In response to the reviewer’s suggestion, we have shortened and thoroughly restructured Section 3.2, Mechanistic Insights into Peritoneal Fibrosis with a Focus on SIRT1. Furthermore, we have added a new subsection, 3.6, SIRT1 in Lipid Metabolism, which addresses the role of SIRT1 in cholesterol and lipid homeostasis, including its interactions with key regulatory proteins and nuclear receptors. This section also discusses the implications of our findings on circulating SIRT1 levels and serum cholesterol concentrations.
These revisions were intended to enhance the clarity, focus, and mechanistic depth of the Discussion while ensuring alignment with the existing literature and the reviewer’s recommendations.
- The conclusion should include: Include relevant findings (cholesterol and phosphates as independent predictors).
Revised as requested.
9. There are some minor grammatical and syntax errors throughout the article and in one of the tables. Please review them.
Thank you for your comment. We have thoroughly reviewed the entire manuscript, including all tables, and corrected all grammatical and syntax errors. Furthermore, we have enhanced the overall language quality to improve clarity, readability, and ensure consistency with academic and professional stand

Reviewer 2 Report
Comments and Suggestions for Authors
First of all, I would like to appreciate to review for this manuscript. This manuscript is written well.
The study addresses a critical area in chronic kidney disease (CKD) and peritoneal dialysis by investigating the role of SIRT1, a relevant biomarker. Including three participant groups (CAPD patients, conservatively treated CKD patients, and healthy controls) strengthens the study's ability to identify contrasts and correlations. Using ELISA to measure serum SIRT1 levels ensures a precise, reliable biochemical evaluation. The study highlights connections between SIRT1 levels and cardiovascular stress, which is highly significant in CKD populations.
However, there are few limitations in this study.
1) With only 165 participants, the study may lack generalizability, especially considering the potential variability in pharmacotherapy and individual patient factors.
2) Without experimental intervention, causal relationships between SIRT1 levels and specific pathophysiological outcomes remain unclear.
3) Although phosphate and cholesterol were identified as predictors, additional metabolic and environmental factors could provide a more comprehensive understanding.
So, if possible, could you add these as a study limitation?
Author Response
RESPONSE TO REVIEWER 2 COMMENTS
We thank the Reviewer for the positive evaluation and constructive suggestions. As recommended, we have expanded the limitations of the study in conclusions section to highlight the relatively small sample size, the observational design, and the need to consider additional metabolic and environmental factors.
However, there are few limitations in this study.
1) With only 165 participants, the study may lack generalizability, especially considering the potential variability in pharmacotherapy and individual patient factors.
2) Without experimental intervention, causal relationships between SIRT1 levels and specific pathophysiological outcomes remain unclear.
3) Although phosphate and cholesterol were identified as predictors, additional metabolic and environmental factors could provide a more comprehensive understanding.
So, if possible, could you add these as a study limitation?
